# How Well Environmental Design Is and Can Be Suited to People with Autism Spectrum Disorder (ASD): A Natural Language Processing Analysis

**DOI:** 10.3390/ijerph19095037

**Published:** 2022-04-21

**Authors:** Moti Zwilling, Beni R. Levy

**Affiliations:** 1Department of Economics and Business Administration, Ariel University, Ariel 40700, Israel; 2School of Architecture, Ariel University, Ariel 40700, Israel; benirl@ariel.ac.il

**Keywords:** evironmental design, space, autism spectrum disorder, natural language processing, adaptive architectural design

## Abstract

The quality of life of people diagnosed as having Autistic Spectrum Disorder (ASD) is essential for increasing their self-reliance and reducing their communication problems in order to allow them to work, take care of themselves, and develop a capacity to intercommunicate with their surroundings. Their need to organize their day-to-day and workplace surroundings has been addressed in the literature via long-term intervention programs aimed to imbue people with ASD with interpersonal communication capabilities. Yet, there is still a gap in the literature regarding new design methods aimed at creating a safe and friendly environment adapted to the needs of people with ASD. Therefore, this study has two objectives: (1) to shed light on the existing factors and methods related to workplaces designed to be friendly to people with ASD, specifically adults, through a natural language processing (NLP) analysis of existing scientific papers in the field of architecture and design; and (2) to explore the factors that might assist in improving the design and architecture of adaptive spaces for people with ASD by analyzing a corpus of experts’ documents. The study findings and their implications are analyzed and discussed.

## 1. Introduction

Autism is a lifelong disability that affects the ability to communicate and to relate to others [1]. All autistic persons share certain difficulties, but their special condition affects them in different ways. While some may live relatively independent lives, others may have special learning and working disabilities [2]. People with autism have difficulties in four main areas: social interaction, communication, repetitive behaviors, and sensory impairment (such as hypoactivity, hyperactivity, and fluctuations between these states), which is considered the most prevalent issue [3,4]. The need to enable autistic people to integrate into the general society, despite their different abilities and special talents, is essential for their welfare in their day-to-day life [5]. It is also important for a sustainable society and can even be perceived as an advantage for any group of people, including potential employers [6]. 

Incorporating autistic children and youth into educational institutions and society in general is a goal that has been achieved quite successfully by paying special attention to the adaptation of the physical surroundings to their special needs [7]. However, not much attention has been given to autistic adults’ environmental requirements in the workplace, which should allow them to work properly within the general work market scenery [8,9,10]. Barriers to employment faced by people with autism spectrum disorder (ASD) include lack of access to all levels of education and vocational training, lack of adequate support during the transition to adult life, and specific obstacles arising throughout the recruitment process such as discrimination in relation to employment and inadequate environmental conditions adjusted to their special needs [5,6,7,11]. People with ASD are unique, also regarding their personalities, characteristics, and abilities [8]. They may have an above-average level of skills in specific areas that may be considered an advantage for employers, such as high levels of concentration, a special ability for repetitive tasks, increased attention to details, specific technical skills (e.g., outstanding record-keeping and problem-solving talents), and pronounced perfectionism, loyalty, and reliability. In fact, according to Cope and Remington [12], people with ASD may reveal cognitive advantages such as superior creativity, memory, and focus, as well as honesty and dedication to their work. However, they suffer from several disadvantages, such as poor integration and the need for vocational support [13]. In order to adapt the work environment to the physical and emotional needs of people with ASD, it should be designed with adjustments regarding noise, lighting, crowding, and orientation in mind [9,10,11,14].

A partial list of jobs considered suitable for people with ASD includes the following: (1) writing tasks [15], appropriate for people who are good with facts and can put a creative sentence together [15,16]; (2) assembly line tasks [16], which require repetitive movements and actions; (3) drafting jobs that involve making detailed drawings from the specifics provided by an architect or engineer [16,17]; (4) paralegal positions, which are defined as drafting documents and organizing files to support the work of lawyers [16,17]; (5) plumbing tasks, which require working with the hands with precision and patience [15]; (6) routine jobs such as translation posts and filing documents, which fit people who speak and read fluently in various languages [16,18]; (7) photography tasks where the individual captures images of wildlife, pets, people, landscapes, and buildings [16]; (8) video and computer programming, which require an artistic talent, technical skills, and storytelling abilities [16,19]; (9) working with animals, either at a zoo or as a pet sitter, a dog walker, or a veterinary technician [16]; and (10) mechanical jobs, especially those that involve taking things apart and putting them back together [16,17,20].

The incorporation of people with ASD into the workplace can indeed be an asset to employers, but it requires adjusting the workplace’s spatial conditions. Reasonable environmental adjustments intended to allow individuals with ASD to participate fully in their job include offering a quiet work setting that induces intense concentration; minimizing noise, lights, smells, and visual disturbances; and providing clear means of orientation in the workplace [21]. The exact requirements and recommendations for a proper ASD-friendly work environment may vary from those intended to accommodate other types of disabilities [22]. 

In this respect, the most important ASD-related adjustments in the workplace’s architectural design involve a number of categories as mentioned, for example, by Cassidy [23,24]: (1) Lighting: given the extreme sensitivity of people with ASD to light and the heat it causes, the use of natural lighting is recommended but requires avoiding direct sunlight. In terms of artificial lighting, LED lights with adjustable intensity lighting systems and a diffuse light source to avoid glare are preferrable to fluorescent lighting (because of the latter’s tendency to flicker and the low humming sound it emits). (2) Materials and textures: it is recommended to avoid environmental rigidity by creating smooth and wide surfaces using a limited number of simple, non-reflective, and robust materials and textures. (3) Colors: the use of soft, natural colors, and the limitation of color contrasts, are preferred; plants can be used to separate spaces devoted to different functions; soft colors should be applied along walking paths, while bright colors should be used only at the nodes. (4) Acoustics: the work environment should be as free of disturbing sounds as possible. To that end, the recommended changes include installing anti-trauma and sound-absorbing flooring, as well as carpeting and wood furniture; adopting sound-reducing techniques regarding the external walls of the building, particularly if they are located in areas adjacent to high noise sources; building soundproof walls and roofs to ensure sound insulation between the rooms; and making use of double-glazed windows. (5) Smells and air quality: this requires ensuring that the space is well ventilated a enjoys a good air conditioning system, as well as avoiding strong smells. (6) Transition spaces: these are used to provide transition elements and areas between different activities and spaces to allow individuals to orient themselves. They involve designing buffer areas, such as gardens and outdoor spaces, and defining a clear spatial distinction between the different activity-designated places by organizing them into compartments, arranging the furniture in specific patterns, and even employing variances in lighting. The recommended changes include avoiding multifunctional and ambiguous areas to reduce sensory confusion and providing calming and soothing areas for one-to-one interaction that allow people to retreat from overwhelming social situations [23,25]. These spaces should be small and neutral in terms of the sensory environment so that they foster minimal distractions. They should be positioned in such a way that allows continuous supervision (from the outside) of the activities taking place in them, while being partially and perceptually separated from the main space through the use of different design solutions. Transition spaces should be well-defined and simple, preferring multifunctional circulation areas to traditional corridors to allow the possibility of choosing the best use of the space. (7) Visual relationships: the workspace should be organized so as to facilitate discreet supervision of the different environments. It should be designed to ensure a good visual relationship between different internal and external environments to allow supervisors to be continuously aware of the situation of employees with ASD and to intervene to avoid any possible distress. (8) Circulation, predictability, and routine: the architectural design should emphasize order, sequences, and routines by organizing activities and functions accordingly, starting in the specific work areas up to the entire building. Visual aids should be positioned so as to provide instructions for activities that can take place in different environments. Corners, which can hide dangers or unexpected situations, should be avoided, and elements of consistency should be included in outdoor environments to create predictable patterns. (9) Circulation: the design should provide opportunities for choosing between different levels of social interaction by including both personal spaces (to accommodate small groups and to enhance feelings of closeness, intimacy, and safety) and collective spaces for work and leisure. This provides a hierarchy of spaces for potential use in different situations, including spacious transition and circulation areas directly connected to the various workspaces that allow a person to easily decide where to go. (10) Proportions: spaces should not be too small or have very high or very low ceilings (the latter convey a sense of oppression). Long corridors that can turn into dead (unutilized) spaces should be avoided. (11) Orientation: this is defined as the adoption of a process of visual supports, such as images, words, or colors, for the purpose of providing information about the use and function of the space and to inform about potentially critical points such as stairs or slopes. The creation of evident paths by using color codes and labels, and the enhancement of visual features through the introduction of colors on floors, walls, and doors, support a person’s spatial orientation. (12) Specificity in functionality: the architectural design and construction of the work environment should pay specific attention to the unique working abilities of employees with ASD, their special sensitivities, and the jobs in which they may excel [9,10,26,27]. 

Although the focus of this study lies mainly with the design and adaptation of workplace environments to people with ASD, the architectural design of educational institutions, such as universities and schools, should also consider the needs of students and faculty with ASD. For example, in both the workplace and school, it is important to make use of sensory elements that help individuals to easily perceive and orient themselves in their surroundings, and also of technologies to reduce destructive noises. Elements nurturing the psychological well-being of people with ASD should also be implemented in both settings, albeit perhaps in different ways. For instance, workplace ergonomics may induce productivity while school ergonomics may be better suited for social integration. Moreover, the adaptation of the workplace’s design to people with ASD is also influenced by the designer’s understanding of the range and severity of their symptoms. People with ASD who are hyposensitive require overstimulation by means of sensory information (lights, crowds, and noises), while those who are hypersensitive need the opposite. In addition, in workplaces, employees with ASD can work in small groups or alone. In contrast, the common architectural approach regarding schools is characterized by the perception that the design of learning spaces should not affect the amount of stimulation provided to students who do not have ASD. This perception is based on the assumption that too many adaptations in the learning environment may not prepare individuals with ASD for the “real world” [28,29]. A case in point refers to the “treatment rooms” built in educational institutions (but not in workplaces) to teach individuals to deal with specific disabilities, such as speech therapy rooms. In the case of students with ASD, the design of such rooms requires special adaptations by means of acoustical elements. However, as Mostafa [28] (p. 204) mentions, in educational environments it is important to “avoid the ‘greenhouse’ effect, where a child becomes dependant upon the optimum acoustical quality of the room and is unable to function and generalize his skills outside of it”.

Another important aspect to be examined as part of the architectural design of workplaces refers to existing guidelines in countries which have specific employment programs for individuals with ASD. For example, Vogeley et al. [30] explored the readiness of the German health and social care systems to provide on-the-job training to people with ASD in their workplace. The authors found that according to Herzberg’s theory [31], inadequate conditions in the workplace may result in such employees becoming dissatisfied with the job. 

Moreover, Wehman et al. [32], as well as Jaarsma and Welin [33], have shown that although vulnerable employees, and people with ASD among them, have a universal right to work [34], employers still perceive environmental factors to be obstacles for the successful performance and integration of individuals with ASD in workplaces. Johnson et al. [35], for example, have examined some of the strengths and weaknesses related to the employment of people with ASD. According to their study, stable technology-oriented companies such as Microsoft, SAP, and JP Morgan have launched special programs for individuals with ASD that may answer to their special needs; however, their employment in most cases was short termed, since employers are not trained and aware enough to understand how to cope with negative behaviors among such employees. This observation is also supported by other studies, e.g., [36].

The literature analysis above shows that the employment-related needs of people with ASD worldwide constitute a real challenge for society and employers alike, and propel the latter to invest great efforts to adapt the workplace to the requirements of such potential employees. It is implicitly assumed that such adaptation is also adequate for non-ASD personnel. 

### Research Aim and Research Questions

In recent years, the academic and professional literatures have dealt largely with the subject of incorporating children and youth with ASD into educational institutions and society in general. A considerable body of literature has been dedicated to the adaptation of the study environment to the needs of youngsters with ASD. However, only a few studies have explored the environmental requirements of adult people with ASD with a view to understanding the workplace conditions that allow them to take part in the general work market [18]. Therefore, the aim of this study is to investigate the relationship between the unique abilities of adult people with ASD and the consequent adaptation of the working conditions for their benefit, as well as for the benefit of the other employees. To do so, the current study proposes to explore the following research questions:In what manner is the environmental and architectural design of workplaces adapted to people with ASD, according to the scientific literature and the expert opinion corpus?What are the environmental needs of people with ASD, and which of their unique attributes may be integrated into the new workplace design?What recommendations can be suggested to companies on the basis of the combination of an adaptable environmental and architectural design with the special needs and unique qualifications of people with ASD?

## 2. Materials and Methods

### 2.1. General

To address the research questions, the authors analyzed the content of scientific and experts’ papers in the inspected domain through the use of two natural language processing approaches (Topic Modelling and Word Cloud) provided by the Orange^TM^ data mining toolkit (Developed by the University of Ljubljana, Ljubljana, Slovenia) as described below. Specifically, the following objective and the ensuing derived questions were constructed: 

**Objective**: Analysis of the relevant scientific and expert opinion literature for the period between 1995 and 2022.

***Derived Question 1***: What trends regarding environmental and architectural design are portrayed in the literature? These trends will be used to understand whether the adjustment of workplace architectural design to people with ASD has improved over the years (Research Question #1).

***Derived Question 2***: What can be learned from the analysis of the extant literature about the ASD-focused environmental and architectural design adopted in the workplace? These findings will be used to better understand the special needs of people with ASD in order to provide them with a supportive environment for learning and working. The definition of unique attributes or adjustments of the new ASD-friendly architectural design may serve as recommendations for architects and designers (Research Questions #2 and #3)

***Derived Question 3***: To what extent have companies addressed the special needs and unique qualification of people with ASD throughout the years by means of an adaptive environmental and architectural design of the workplace? The analysis of the companies’ attitudes according to the expert corpus’ content may contribute to understand the progress (if any) of the design of work and educational spaces over the years regarding its adaptation to the environmental and special needs of people with ASD.

### 2.2. Data Collection

For the purpose of tracing the existing scientific literature and answering the research questions derived from the above-mentioned objective, we conducted two processes. The first involved collecting a corpus of 924 scientific articles from the Web of Science (WoS) database. The corpus was based on scientific papers published in peer-reviewed journals and conference proceedings identified on the basis of the following keywords, each considered separately: “Autism”, “Environment”, and “Design”. A further specification of the explored domain using the words “Environmental Design” and “Autism” yielded 389 articles, while a more refined search with the words “Environmental Design” and “People with Autism” obtained only 42 results. A deeper filtering of articles in the inspected domain using a conjunction of the words “Architecture for Autism” and “Built Environment” as topics provided 10 scientific articles that represent the inspected scientific domain. In addition, 10 selected articles by experts focused on “Architectural design of workplaces appropriate for autistic people” were downloaded from the Internet for further analysis. It should be noted that the number of documents provided by experts in this domain was found to be quite limited. The contents of the scientific and the experts’ documents were analyzed according to Sardi et al. [37] to verify their compliance with each of the following criteria: (1) do they related to the topic of architecture and design? and (2) are they written in English? Finally, to validate the expert document contents’ relationship with the explored topic (“Architectural Design for Autistic People”), the title and content of all 10 expert opinion articles were fully read. The same process was applied to the full list of papers collected from the scientific corpus (a total of 10 articles). Additional information about the various Web of Science categories, the number of records, and the percentage of documents from all the explored domains appears in Table 1. 

## 3. Methodology

### 3.1. Data Analysis: Extraction of Scientific and Experts’ Papers According to Topic and Affiliated Keywords 

Scientific papers were extracted from the Web of Science (WoS) database using the InCites graphical interface. Following the guidelines of Gheyas and Abdallah [38], a systematic literature corpus collection was performed for the period between 1995 and 2022. The search was configured by the graphical search combo box and contained the following editions:Science Citation Index Expanded (SCI-EXPANDED):1993–present;Social Sciences Citation Index (SSCI): 1993–present;Arts and Humanities Citation Index (AHCI): 1993–present;Conference Proceedings Citation Index–Science (CPCI-S): 1990–present;Conference Proceedings Citation Index—Social Science and Humanities (CPCI-SSH): 1990–present;Emerging Sources Citations Index (ESCI): 2015–present.

The papers’ titles were searched for topics using keywords suggested by Sardi et al. [37]: “Architecture” and/or “Design” and/or “Environmental Design” and/or “Built Environment” and/or “Working Places” and/or “Working Environments” and/or “Autism” and/or “Autistic People” and/or “for Autism”. In addition, since the concept of universal design is also common in special education architecture, the following keywords were searched together: “Universal Design”, “Autism”, and “Workplace”. The experts’ articles were searched through the Google^TM^ search engine, where the following words were used: “Architecture” and/or “Design” and/or “Environmental Design” and/or “Autism” and/or “Autism People”.

The contents of the articles’ abstracts were analyzed using the Orange^TM^ toolkit (Figure 1). The toolkit enables the design and easy implementation of NLP text analysis models, such as topic modeling and bag of words, used for the construction of word cloud diagrams. 

### 3.2. Topic Modeling and Word Cloud

Topic models, such as the Latent Dirichlet Allocation (LDA) technique, are considered to be unsupervised classification methods used to find correlations between topics, changes of topics over time, and the semantic understanding of short texts based on prominent word co-occurrence information [39,40]. To obtain the prominent topics or prominent words within the inspected documents, topic modeling was implemented through the Orange^TM^ toolkit. It was used to analyze and find the frequency of words in the articles’ abstracts, thus providing an indication about relevance to the inspected domain. Following Korenčcić et al. [41], the LDA unsupervised learning algorithm, which is one of the most popular topic modeling methods, was used. The algorithm is designed to analyze a large volume of unlabeled text. This analysis was conducted through the evaluation of clusters of words that frequently appear together [42,43]. Assuming that similar words may appear in similar contexts, the conclusion is that finding the same or similar words in different texts leads to the same topic or indicates the word’s importance. 

The word cloud technique is a method of text data analysis used on mainly unstructured text data that provides a simple visualization of frequent words in a corpus. The cloud is based on the most frequent words within the corpus, which are presented in a spatial layout where the font sizes of the words are proportional to their relevance or frequency [44,45,46].

These methods were implemented in the following manner: (1) Two separate corpora of documents were constructed (based on scientific articles and expert opinion columns). (2) The text of each corpus was preprocessed by the preprocess tool to filter white spaces and punctuation. (3) Topic modeling and word cloud components were adjusted to each clean corpus element. In addition, the sentiment analysis component was used to evaluate whether the text contained negative, positive, or neutral attitudes, and the heat map component was used to support it with visualization attribute values.

## 4. Results

### 4.1. Scientific Trends in the WoS Repository 

The documents’ distribution for the 1995–2022 period revealed a consistent trend in the number of citations dealing with autism and environmental design when “Autism”, “Environment”, and “Design” were used as separate words (Figure 2). However, in the years 2016–2022, specific articles in the domain of autism and workplace design and/or built design became more prominent. Hence, when the search was conducted with the words “Autism” and “Environmental Design”, a total of 364 publications with 11,739 citations and 41.17 average citations per item was obtained. After excluding the irrelevant papers from the obtained repository and refining the search using the sentences “Architecture for Autism” and/or “Built Environment”, only 10 papers for the period of 2016–2022 remained (Table 2). These papers had 31 citations with 3.1 average citations per item. The data visualization report showed that these 10 papers were affiliated with the following domains: the domain of Public Environmental Occupational Health included 3 papers; the domain of Architecture contained 2 papers; and the domains of Environmental Studies, Special Education (provided by WoS as Education Special), Engineering–Multidisciplinary, Environmental Sciences, Humanities–Multidisciplinary, Regional Urban Planning, Urban Studies, and Rehabilitation comprised 1 paper each (Figure 3). It should be noted that some of these 10 papers belonged to more than one domain. An evaluation of the papers’ titles showed that they dealt with the topics of inclusive space design, acoustical design, friendly and non-noisy workspaces, and lighting conditions. A scoping review following Tola et al. [27], supported by the other papers, showed that acoustic, smell, noise, and light stimuli, along with adequate visualization and friendly environmental design, were viewed as the main factors to be taken into consideration in the architectural design of workplaces for people with ASD. In addition, the article by Sheppard-Jones et al. [47] provides guiding principles of universal design for vocational rehabilitation that facilitate the fulfillment of the needs of people with ASD on the march toward more productive workplaces. Moreover, universal design was mentioned as an important factor used to support students with ASD to engage with other students and acquire sufficient training, social skills, and readiness for the job market [48].

### 4.2. Analysis of the Experts’ Repository

This section describes the analysis of the experts’ data according to the primary categories derived from the expert opinion papers. As Figure 4 indicates, these categories relate primarily to the minimization of sources of sensory sensitivity and visual disturbances (such as noise and lighting), and to space planning designed to provide friendly spaces where co-workers and employees with ASD can communicate in a discrete and comfortable manner. 

To emphasize and support these categories, the following text evolves from the articles:4.The Center for Autism Research and The Children’s Hospital of Philadelphia (2020) [49] suggest the following autism friendly design ideas:

#### 4.2.1. Sound Issues

Excess noise can be at best distracting and at worst sensory overload.Care should be taken to reduce sounds, rattles, and related noises from ventilation and related systems.Insulation from other noises in the environment (traffic, for example) is important for similar reasons.Plan to use an acoustic ceiling to reduce noise, as well as carpet, which will help absorb sound.

#### 4.2.2. Lighting


*Indirect lighting other than fluorescent lighting should be used because fluorescent lighting has a flicker and hum, which, though imperceptible to many, may be uncomfortable to the sensory sensitivities of many individuals on the autism spectrum. […]*


#### 4.2.3. Space Planning

Divide areas for certain activities, where possible, and segregate by color, etc. (through use of floor finishes, different colored shelving, etc.).Be sure to incorporate “quiet rooms” within the building, which are spaces where a family can retreat when a child begins to get overwhelmed. Ideally, they should be located throughout the building so a family does not have to go a long distance to find one.The space within the building should be easy to navigate to reduce confusion in how to get to different areas within the building.

Neal Catalano writes in his article “Workplace Challenges for Individuals with Autism” (2020) published on the Spectrum of Hope website: “*People with ASD can also work with their environment, coworkers, and employers to minimize sources of sensory sensitivity. These may include loud noises or distractions, which may prove stressful, anxiety-inducing, and disruptive to productivity. Some people with autism experience sensory meltdowns as a result of stress, fear, or overstimulation. [,,,] There is no one-size-fits-all solution for overcoming workplace challenges with autism, but there are numerous helpful strategies*” [50].The article on the Findatopdoc website titled: “Adults with Autism in the Workplace: Accommodation Tips” (2020) include the following advice: “***Stress management****. Allow an option to go to a quiet place a few times a day (or as needed). Not every job has a cubicle. Allow noise-reducing or eliminating headphones* [51]. *[…]* ***Relocation of workspace where possible****. Reduction or elimination of loud noises, fluorescent lights, or a hectic, highly trafficked area. Reduction of stress is important*”.Sherrell, in her 2021 paper “What to know about autism discrimination in the workplace” published on the Medical News Today website, says that reasonable adjustments include: “*Offering a quiet working place or a “do not disturb” sign that individuals can use when they require intense concentration. […] Minimizing noise, lights, and visual disturbances by using desk partitions, low lights, and providing noise-cancelling headphones*” [52].Pacelli, in her 2014 paper “How To Build An Autism-Friendly Workplace” published on the Fast Company website^4^, says: “We’ve seen first-hand how an autism-friendly workplace contributes to a more effective and balanced workplace. It’s incumbent on today’s leaders to create an environment where employers and autistic employees not just survive, but thrive” [53].

### 4.3. Word Cloud Analysis and Topic Modeling

Figure 5 and Figure 6 show the scientific literature and the experts’ discourse on the importance of environmental design for people with ASD and its contribution to their quality of life and integration into society. Both the scientific and the experts’ papers mention the importance of “adaptive” designs that may improve the quality of life of people with ASD and especially their integration into workplaces. For example, words such as “Building, “Urbanisms”, “Future Architecture”, “Sensory”, “Improving”, “Spaces”, “Design”, and “Intervention” are frequently found in these word clouds, which points to their importance. Overall, the word cloud diagram shows that while scientific documents focus mainly on the theoretical impact of environmental design (derived from the words “Autism”, “Results”, “Study”, “Training”, “Conclusions”, “Methods”, and “Research”, which are prominent in the scientific papers’ word cloud distribution (Figure 5), experts’ documents focus mostly on the implementation of adapted and effective architectural designs for people with ASD as part of an intervention programs aimed to integrate them into workplaces (as seen, for example, in the words “Design”, “Spaces”, “Friendly”, and “Sensory”, which are prominent in these articles’ word cloud distribution (Figure 6)). Moreover, topic modeling shows that the design of sensory and urban environments may improve the adaptation of people with ASD to their workplace surroundings. They are also important in mitigating the communication difficulties of persons with ASD and contribute to their future development as self-reliant individuals. These findings can be inferred from words such as “Sensory”, “Interaction”, “Architecture”, “Urbanism”, “Universal”, and “Autism”, which are prominent in the expert opinion corpus (Figure 7). Lastly, sentiment analysis and heat map components point to a positive attitude on the part of architectural experts regarding the adjustment of workplace design to people with ASD. 

## 5. Discussion

The findings show that the design of workplaces should be adapted to people with ASD. This is of great importance to the latter’s possibility to contribute to their employers and to society. This may help them feel equal among “typical” people by opening the door for them to be integrated into their society as contributors to the local workforce. Although the inclusion of people with ASD in the workplace might be considered a complex and almost unreachable objective, the scientific literature and the analysis of architectural experts show that a rational environmental design that considers their needs and obstacles in the workplace in conjunction with those of typical employees may be beneficial to the individuals with ASD, their employers, and their co-workers. 

The design of workplace environments should consider the possible obstacles related to the human environment, the job requirements, and also physical and sensory factors [54,55]. For example, it is suggested that employers adjust the competitive work environment to the needs of the employees with ASD by adding safety and orientation-enabling accessories and making it friendly enough for people with ASD who want to engage with their soundings. These elements should be selected based on the understanding of sensory and physical difficulties that affect personnel with ASD. In addition, difficulties related to social communication such as social immaturity and social naïveté, which are perceived as inappropriate social behaviors, additional comorbidities, and challenging behaviors, are also considered to be factors that may hinder the successful integration of employees with ASD in workplaces supposedly adapted to their needs. Tomczak [10] suggests that the environmental design should also incorporate smart technology suitable for several of the adaptation objectives, such as improving the interpersonal communication of employees with ASD with their surroundings through chatbots and communication devices or monitoring their level of stress through wearable devices. The author believes that turning the workplace into a smart environment improves the productivity of people with ASD and their feeling of belonging to the firm. 

The results of the text analysis of scientific and expert opinion documents as well as the word cloud analysis show that people with ASD need a friendly climate that takes into consideration sensory stimuli and communication difficulties [10]. The adapted environmental design is intended to help them engage with other employees and maximize their productivity. Examples of architectural design sketches of appropriate work environments adapted specifically, but not uniquely, to individuals with ASD are shown in Figure 8, Figure 9, Figure 10 and Figure 11 (all sketches were drawn by one of the authors). These sketches demonstrate several solutions regarding the proper spatial planning and avoidance of sensory disturbances that are expected to contribute to create an ASD-friendly workplace. Figure 8 illustrates a design that includes relaxation spaces, acoustic ceilings and carpets, visual clearance, clear and simple circulation paths, and different well-defined work and rest areas. Figure 9 portrays the design of a friendly personal workspace within a wider open-space arrangement that incorporates natural and artificial lighting and acoustic panels. Figure 10 shows a design that comprises intimate spaces with artificial ventilation and air conditioning, non-fragrant vegetation in open partitions, and various peaceful colors defining different areas. Figure 11 shows a design that includes controlled daylight, walls with images of pastoral views, and furniture arrangements that create a homey atmosphere.

Our findings provide a number of important contributions to the existing literature in the respective domain. (1) This study provides a framework for selecting a list of factors related to architectural design in work and/or educational buildings to be adapted to the needs of people with ASD. (2) It provides a list of professional adjustments that may assist architects to improve their designs for and adapt them to the special needs of people with ASD in workplaces and educational environments, such as accommodating sensory-related accessories in the soundings in a way that improves the productivity or educational skills of people with ASD. These professional adjustments arise from the analysis of the expert opinions’ corpus, and they include specific instructions such as reducing the sounds emitted by the ventilation system or using acoustic ceilings and carpets to absorb undesirable noises. (3) The experts analysis findings also show that adapting the design of the workplace to people with ASD will influence the employee productivity as well as reduce management problems. For instance, reducing the stress of employees with ASD improves their personal communication skills and helps them integrate better with co-workers and with their tasks. (4) Finally, this study’s findings and the affiliated literature help architecture practitioners achieve a detailed understanding of the special needs and obstacles of people with ASD, thus providing them with an opportunity to design environments that better address the latter’s needs and ultimately to fully integrate them into society and improve their quality of life.

## 6. Limitations and Future Research

This research is subject to some potential limitations. First, the study data were extracted from the scientific literature and expert opinion articles, and no analyses were conducted based on closed and open questionnaires. Future studies should consider conducting interviews with employers and employees with ASD already working in adapted and non-adapted workplaces, and analyzing their findings using quantitative and qualitative (e.g., content analysis) tools. These data may assist in supporting and strengthening the current study’s recommendations. We suggest that the future questionnaire includes the architectural sketches proposed in this study, to be shown to the respondents (both employers and autistic working adults with ASD). The questionnaire may thus explore the impact of the suggested designs on the adaptation of people with ASD to their workplace as well as the ability of potential employers to improve the communication and commitment such employees. In addition, pictures of exiting architectural designs of buildings adapted for people with ASD (which have not been used in this study) should also be included in the questionnaire. A follow up study should incorporate pictures and images that may better assist practitioners to design a building space suited to people with ASD people in work and educational institutions. 

Lastly, the data search was mainly performed on the Web of Science core collection, which is considered to be one of the most prominent academic search engines that allows exporting full scientific papers from several sources. However, future studies should explore also additional repositories such as Google Scholar (which in most cases is limited to abstracts only). Such analysis might provide additional information that may expand and improve the existing corpora of articles (extracted from the WoS) and strengthen the present study’s findings.

## 7. Conclusions

In general, the above-mentioned findings give rise to the following principles regarding an adequate workplace design suited to ASD people with special emphasis on sensory sensitivities (lights, noise, and smells) and spatial planning (crowding).


*Lighting Conditions:*
Making use of natural light; avoiding direct sunlight and glare; providing shaded spaces;Using a limited number of simple and non-reflective materials and textures;Employing soft, natural colors and limiting color contrasts.



*Acoustic Conditions:*
Creating sound-absorbing flooring, ceilings, and walls using carpeting, wood, and soft fabric furniture;Guaranteeing good ventilation and proper air conditioning.



*Olfactory Conditions:*
Using fragrance-free plants to reduce the negative effect of fragrant plants on people with ASD [56,57,58].



*Spaces/Crowding Conditions:*
Using plants to separate environments devoted to different functions;Providing transitional elements and areas between different activities and spaces to allow individuals to orient themselves;Defining clear spatial distinctions between different activity places by means of furniture arrangements and through variances in ceilings and floors;Avoiding multifunctional and ambiguous areas to reduce sensory confusion;Creating calming and soothing areas for one-to-one interactions to provide the possibility to retreat from overwhelming social situations;Keeping a visual relationship with the workspace of employees with ASD to allow the latter to be in touch with the main workspace and also to allow a supervisor to check on the employees in an inobtrusive manner and follow up on their situation;Defining spaces in a simple yet precise manner, preferring visually open circulation areas to traditional corridors to allow the possibility of choosing the best use of the space;Providing opportunities for choosing to participate in different levels of social interaction by including spaces to accommodate small groups that enhance a feeling of closeness, intimacy, and safety;
5.Providing a hierarchy of spaces directly connected to a spacious circulation area that provides the possibility of deciding where to go with ease;
Designing spaces with good proportions, neither too small nor too wide, avoiding low ceilings that convey a sense of oppression.


Taking into consideration all of the above, our conclusions are as follows: the absorption and integration of autistic people in workplaces is considered to be a very important objective of the general society. Although autistic people have many difficulties including low communication skills and sensory sensitivities, the current study shows that with the help of an architectural design adjusted to best suit their needs, it is possible to integrate them into workplaces. This will allow autistic people to feel that they contribute to their surroundings while earning their livelihood independently rather than being a burden to society; on the other hand, employers will be able to make use of the unique attributes that allow ASD people to excel in certain areas compared to the performance of typical employees. Such attributes include their ability to concentrate on details, their different way of thinking and unique form of creativity, and their steadfast loyalty to their workplace. Future studies may expand and strengthen the current study’s implications.

## Figures and Tables

**Figure 1 ijerph-19-05037-f001:**
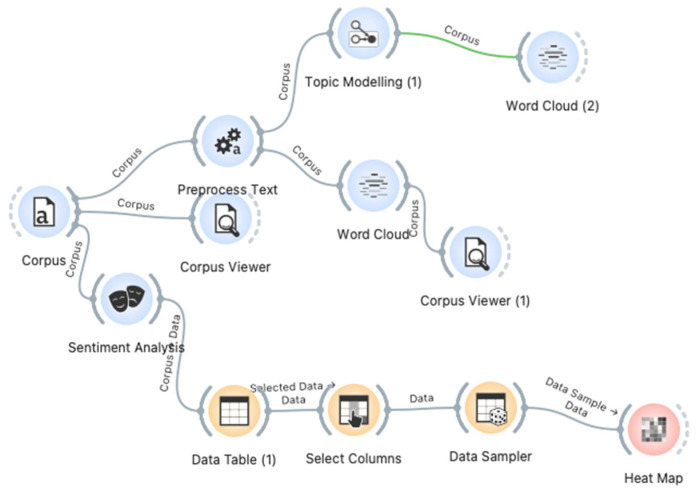
The model created using the Orange^TM^ toolkit.

**Figure 2 ijerph-19-05037-f002:**
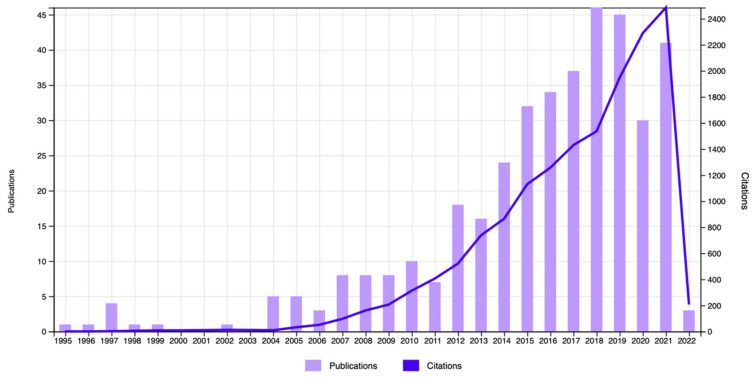
Distribution of articles in the domain of “Environemental Design” and “Autism” for the years 2001–2022.

**Figure 3 ijerph-19-05037-f003:**
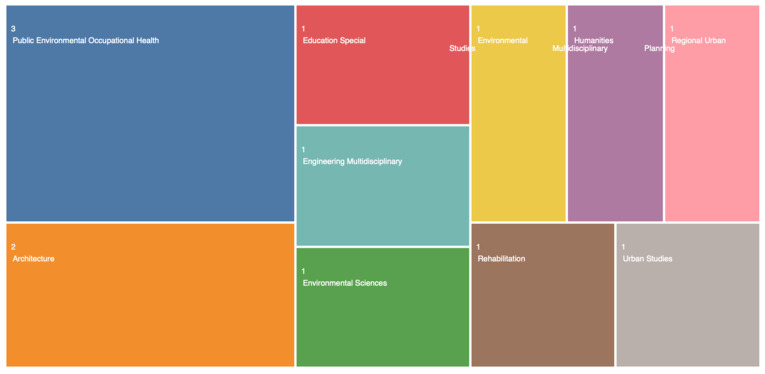
Data visualization scheme based on manuscript relevancy.

**Figure 4 ijerph-19-05037-f004:**
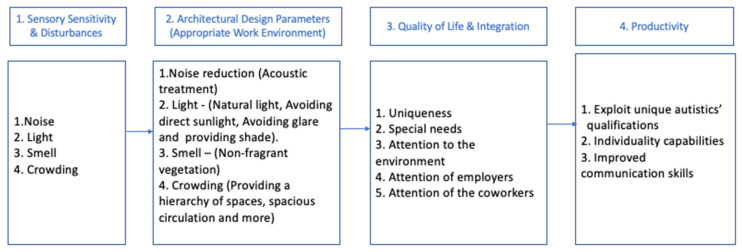
Architectural design topics, requirements, and their contribution to the integration of autistic people in workplaces.

**Figure 5 ijerph-19-05037-f005:**
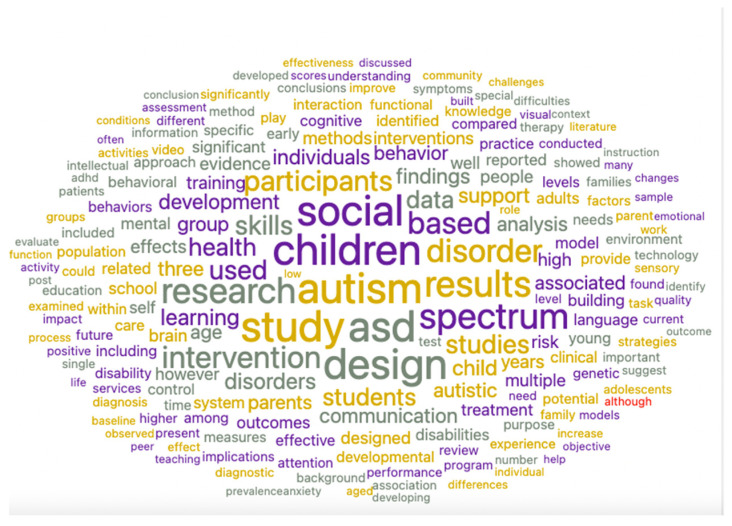
Word cloud distribution of scientific (WoS) documents.

**Figure 6 ijerph-19-05037-f006:**
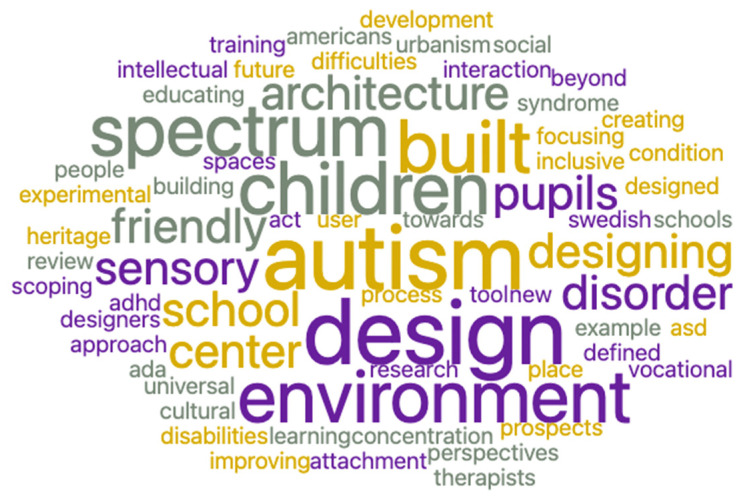
Word cloud distribution of experts’ documents.

**Figure 7 ijerph-19-05037-f007:**
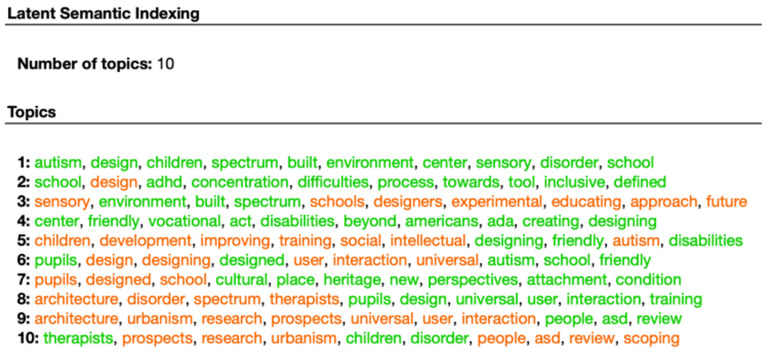
Topic modeling related to 10 scientific documents.

**Figure 8 ijerph-19-05037-f008:**
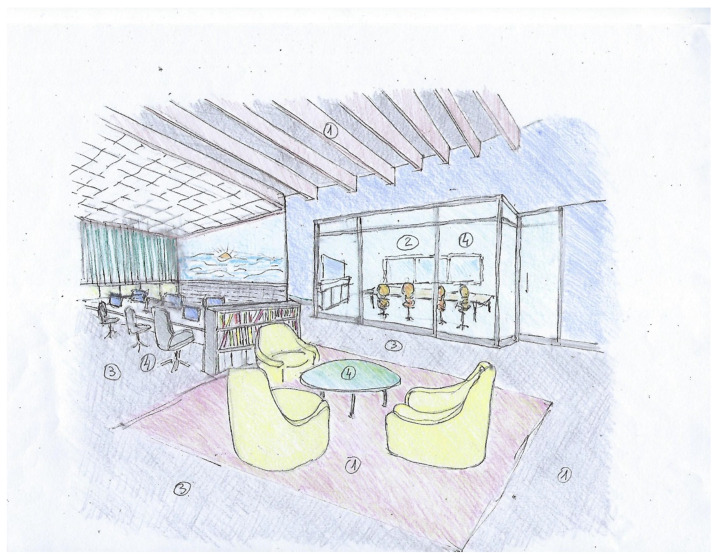
Sketch of an architectural design that illustrates relaxation spaces with acoustic carpets and ceilings ^①^, visual clearance ^②^, clear and simple circulation ^③^, and different well-defined rest and work areas ^④^.

**Figure 9 ijerph-19-05037-f009:**
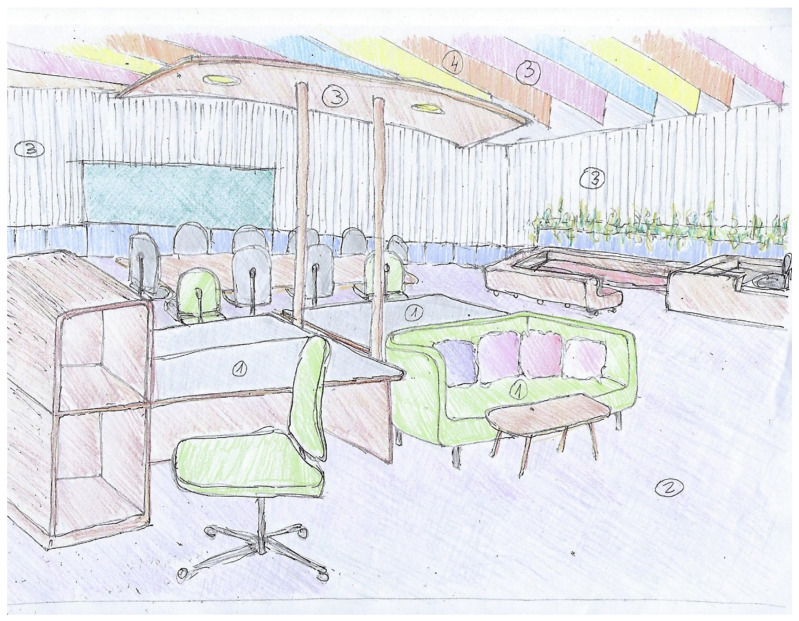
Sketch illustrating a friendly personal workspace ^①^, a wide open space ^②^, natural and artificial lighting ^③^, and acoustic panels ^④^.

**Figure 10 ijerph-19-05037-f010:**
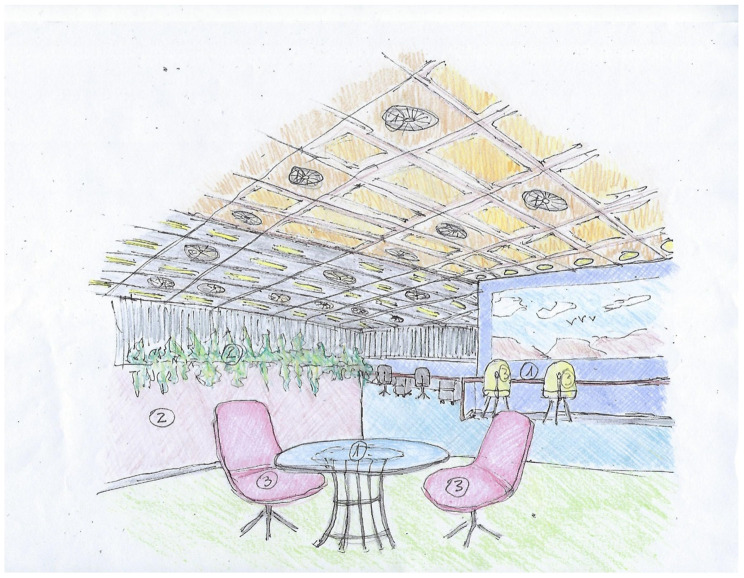
Sketch of an architectural design that illustrates intimate spaces with artificial ventilation and air conditioning ^①^, non-fragrant vegetation in open partitions ^②^, and various peaceful colors used to define different areas ^③^.

**Figure 11 ijerph-19-05037-f011:**
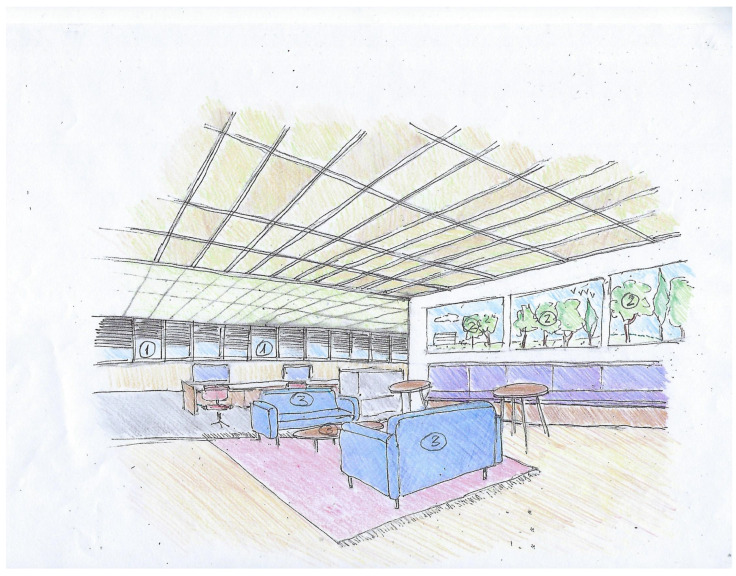
Sketch of an architectural design illustrating the use of controlled daylight ^①^, walls with images of pastoral views ^②^, and furniture arrangements to create a homey atmosphere ^③^.

**Table 1 ijerph-19-05037-t001:** WoS categories (some articles belong to more than one category; see, for example, the Public Environmental Occupational Health domain).

#	Web of Science Categories	Record Count	% of All Domains
1	Engineering Multidisciplinary	1	7.7
2	Educational Special	1	7.7
3	Public Environmental Occupational Health	3	23.07
4	Architecture	1	7.7
5	Environmental Sciences	1	7.7
6	Rehabilitation	1	7.7
7	Humanities Multidisciplinary	1	7.7
8	Regional Urban Planning	1	7.7
9	Urban Studies	1	7.7
10	Environmental Studies	1	7.7

**Table 2 ijerph-19-05037-t002:** WoS papers published in the years 2014–2022.

#	Publication Year	Publication Type	Authors	Author Full Names	Citations	Title	Main Findings
1	2009	J	Tufvesson, C; Tufvesson, J	Tufvesson, Catrin; Tufvesson, Joel	50	The Building Process as a Tool towards an All-Inclusive School. A Swedish Example Focusing on Children with Defined Concentration Difficulties such as ADHD	Environmental factors such as spatial layout, capacity, and function were found to influence the building process. The latter impacts the physical and psychological needs of the users of the building.
2	2016	J	McAllister, K; Sloan, S	McAllister, Keith; Sloan, Sean	31	Designed by the Pupils, for the Pupils: an Autism-Friendly School	Promoting an autism-friendly environment, the article introduces some of the challenges faced by people with ASC in trying to cope with their surroundings, before proceeding to outline the development of a simple school design ‘jigsaw’ kit that helped pupils with ASC to communicate ideas for their perfect school.
3	2016	C	Dalton, C	Dalton, Cathy	6	Interaction Design in the Built Environment: Designing for the ‘Universal User’	The paper explores the principles underlying the design of rooms adapted to ASD people. The paper shows that interactive sensory rooms assist children with ASD to relax and reduce their existing anxiety.
4	2018	J	Love, J	Love, Joan	14	Sensory Spaces: Sensory Learning–an Experimental Approach to Educating our Future Designers to Design Autism Schools	The paper shows that the proper design of educational rooms for special populations such as students with ASD demands an interaction between the students and the educational institution.
5	2019	J	Salama, AM	Salama, Ashraf M.	1	Knowledge Spaces in Architecture and Urbanism–a Preliminary Five-Year Chronicle	The study shows that there are two broad categories for the design of spaces: established and evolving.
6	2020	J	Schofield, J; Scott, C; Spikins, P; Wright, B	Schofield, John; Scott, Callum; Spikins, Penny; Wright, Barry	6	Autism Spectrum Condition and the Built Environment: New Perspectives on Place Attachment and Cultural Heritage	The study provides various perspectives related to the design of building and places perceived by ASD and neurotypical people from the heritage values.
7	2020	J	Clouse, JR; Wood-Nartker, J; Rice, FA	Clouse, Joslin R.; Wood-Nartker, Jeanneane; Rice, Franklyn A.	3	Designing beyond the Americans with Disabilities Act (ADA): Creating an Autism-Friendly Vocational Center	The paper provides ways for the effective design of vocational centers, and also highlights environmental features that are suited to people with ASD.
8	2021	J	Tola, G; Talu, V; Congiu, T; Bain, P; Lindert, J	Tola, Giulia; Talu, Valentina; Congiu, Tanja; Bain, Paul; Lindert, Jutta	2	Built Environment Design and People with Autism Spectrum Disorder (ASD): A Scoping Review	The paper provides three main factors which may be considered during the process of architectural design for people with ASD—sensory quality, intelligibility, and predictability factors.
9	2021	J	Jalalian, H	Jalalian, H.	0	Improving the Intellectual and Social Development of Children with Autism: Design of a Training Center for Autism	The paper provides a platform for the design of a training center used not only for intellectual development but also to development of, social interactions.
10	2021	J	Norozi, N.	Garza, CM.	1	Architecture for Children with Autism Spectrum Disorder and Their Therapists	The paper provides a specific framework for the design of therapy rooms for ASD people.

## Data Availability

The data presented in this study are available on request from the corresponding author.

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
