# Peer review of "How Well Environmental Design Is and Can Be Suited to People with Autism Spectrum Disorder (ASD): A Natural Language Processing Analysis"

_ijerph, 2022, doi:10.3390/ijerph19095037_

Round 1

Reviewer 1 Report

There is a need to show the similarity and difference between a workplace and a school.  Many studies have found that a school can be used as a substitute for a workplace- school bullying and workplace harassment, etc.  Furthermore, what are the current efforts (including the strengths and weaknesses) dealing with ASD in a workplace as a few countries require or recommend the integration of ASD into a workplace?

The discussion needs to address three different issues in a workplace- safe, friendly, and adaptive.  I feel that the discussion is very general and does not focus on these three aspects specified by the authors.  Be careful that safety can include the issues such as bullying (i.e., psychological safety).  Friendliness should blend the issues relating to feeling of engagement in a workplace.  For the term adaptive, it is unclear whether the authors concentrate on a workplace being adaptive to ASD workers or how well ASD workers adapts to a new environment (from a school to a workplace environment).

Author Response

Reply to Reviewer 1's Comments

Revision of “How Well Environmental Design Is and Can Be Suited to People with Autism Spectrum Disorder (ASD): An NLP analysis”

(Ref: IJERPH, Manuscript ID -1657007)

We are grateful to Reviewer 1 for his/her careful reading of our paper and his/her detailed comments. Below are the Reviewer’s comments followed by our responses.

  1. There is a need to show the similarity and difference between a workplace and a school. Many studies have found that a school can be used as a substitute for a workplace- school bullying and workplace harassment, etc.

Reply: Thank you for your insightful comment. Indeed, there are several similarities and differences between workplaces and educational institutions regarding the inclusion of individuals with ASD. Although the focus of the current study is mainly on workplaces, we have added a comparison paragraph that highlights some of the elements shared by both workplaces and schools or unique to each of them. (Refer to lines 215–235.)

  1. Furthermore, what are the current efforts (including the strengths and weaknesses) dealing with ASD in a workplace as a few countries require or recommend the integration of ASD into a workplace?

Reply: We thank you for this question. There are many countries making efforts (with their respective strengths and weaknesses) to influence the willingness of employers to comply with the recommendation for integration of people with ASD in workplaces. To address this comment, we have included the main conclusions of several studies that show the efforts made by employers in different countries to integrate people with ASD in the workplace. We hope that this addition answers the Reviewers’ intent. (Refer to lines 236–249.)

  1. The discussion needs to address three different issues in a workplace- safe, friendly, and adaptive. I feel that the discussion is very general and does not focus on these three aspects specified by the authors. Be careful that safety can include the issues such as bullying (i.e., psychological safety). Friendliness should blend the issues relating to feeling of engagement in a workplace.

Reply: Thank you for pointing this out. The Discussion was elaborated further, providing insights related to the three different issues in workplaces that the Reviewer mentioned as well as additional factors derived from the latest literature.  (Refer to lines 1049–1062.)

  1. For the term adaptive, it is unclear whether the authors concentrate on a workplace being adaptive to ASD workers or how well ASD workers adapts to a new environment (from a school to a workplace environment).

Reply: We highly appreciate this comment. The term “adaptive” was used here to refer to the adaptation of the workplace to employees with ASD The adaptation of workers with ASD to new environments is another interesting and important issue, but one that does not fall within the scope of this manuscript. To address your comment, we emphasized the meaning of the term in lines 20, 215, 222, 251, 589, 1041 and 1224.

Reviewer 2 Report

Line 32: Although hyper-sensitivity is the most recurring perception issue in people with autism, I would suggest to refer more generally to sensory issues, to include those who experience hypo-sensitivity or fluctuations between the two conditions.

Lines 52-64: It would be useful to indicate bibliographic references to support these statements.

Line 353: I would suggest to use the term "typical" (or similar) instead of "regular".

Figures 8-11: The author should specify whether the sketches were made by himself/herself or not. Furthermore, I am not sure that the sketches are really useful for the purpose of representing the spatial requirements that the author intends to show (for example, in the Image 8 circulation space is not adequatelu represented). If the author finds it useful to use images, perhaps he/she could insert some photographs of autism-firendly buildings in which the spatial requirements of interest have been applied (for example: https://www.ga-architects.com/design-autism).

Author Response

Reply to Reviewer 2's Comments

Revision of “How Well Environmental Design Is and Can Be Suited to People with Autism Spectrum Disorder (ASD): An NLP analysis”

(Ref: IJERPH, Manuscript ID -1657007)

We are grateful to Reviewer 2 for his/her careful reading of my paper and his/her detailed comments. Below are the Reviewer’s comments followed by our responses.

  1. Line 32: Although hyper-sensitivity is the most recurring perception issue in people with autism, I would suggest to refer more generally to sensory issues, to include those who experience hypo-sensitivity or fluctuations between the two conditions.

Reply: We highly appreciate this comment. The text has been altered to include the three types of sensory impairment mentioned by the Reviewer (hyposensitivity, hypersensitivity, and fluctuations between the two).  Refer to line 36.

  1. Lines 52-64: It would be useful to indicate bibliographic references to support these statements.

Reply: Thank you for your comment. Bibliographic references were added to support the relevant statements.

  1. Line 353: I would suggest to use the term "typical" (or similar) instead of "regular".

Reply: Thank you for your input. The term “regular” was replaced with the term “typical”. Refer to lines 1043, 1047, and 1424.

  1. Figures 8-11: The author should specify whether the sketches were made by himself/herself or not. Furthermore, I am not sure that the sketches are really useful for the purpose of representing the spatial requirements that the author intends to show (for example, in the Image 8 circulation space is not adequately represented). If the author finds it useful to use images, perhaps he/she could insert some photographs of autism-friendly buildings in which the spatial requirements of interest have been applied (for example: https://www.ga-architects.com/design-autism)

Reply: We highly appreciate this comment. The sketches were produced by the one of the authors (refer to lines 615 and 658). The images were also provided with numbered labels to assist in understanding the description of each illustration (refer to Figures 8–11). The authors did not incorporate any photograph that might be subject to copyrights. All the images are original and prepared by the authors. In future papers, it might be useful to take pictures of ASD-friendly buildings to demonstrate the architectural design illustrated by Figures 8–11. We address this issue in the Limitations section (refer to lines 1449–1452).

Reviewer 3 Report

This study utilized NLP analysis to investigate the relationship between the unique abilities of adult people with ASD and the consequent adaptation of the working conditions for their benefit as well as for the benefit of the other employees. This topic is interesting and important. However, the writing is confusing, and the research methodology is not rigorous enough.

  1. The abbreviation “NLP” is used in the study title. It is better to use the full term “natural language processing” in the title. Moreover, the authors mentioned “machine learning tools” in the title; but it seems only one text mining tool (Orange Toolkit) was used. Are there any other tools used?
  2. The author used “people with autism” in some sentences (e.g., Line 41), “people with autism spectrum disorder” in Line 65, and “individuals with ASD” in Line 68 and 359. The term “autism” was changed to “autism spectrum disorder” by the American Psychiatric Association in 2013. The author should consistently use the term “autism spectrum disorder” or “ASD” in the manuscript.
  3. References are needed for Line 46-49.
  4. In the introduction, paragraph 4, the authors listed a partial list of jobs for people with ASD. However, I did not see references for each of the listed jobs. The authors should make it clear how this list was developed.
  5. Similarly, the authors should explain how the workplace architectural design categories (mentioned in Line 76-132) were developed.
  6. The last paragraph of the introduction, Line 154-159, is unnecessary.
  7. The authors proposed three “research questions” and constructed three “derived questions.” How the “research questions” and the “derived questions” are related?
  8. Please explain why it is appropriate to include a review article (Tola et al. Line 259)? Will this contaminate the results?
  9. Citations are needed for the 10 papers in Table 2. It is better to include a summary of each paper’s main findings in Table 2.
  10. It is good to use the Google search engine as an additional literature searching source, but it is not rigorous enough to use it as a primary source; other databases need to be searched.
  11. How those categories mentioned in section 4.2 were derived from the literature?
  12. What contents in the identified literature were used for NLP analysis? The whole paper or only the paper’s results/findings. The data analysis method needs further description.
  13. The authors already listed many important ASD-related adjustments in the workplace’s architectural design in the introduction. What are the new findings in this study? They should be highlighted in the discussion.
  14. There are a few formatting errors, such as Line 41-52, and Line 67.

Author Response

Reply to Reviewer 3's Comments

Revision of “How Well Environmental Design Is and Can Be Suited to People with Autism Spectrum Disorder (ASD): An NLP analysis”

(Ref: IJERPH, Manuscript ID -1657007)

We are grateful to Reviewer 3 for his/her careful reading of my paper and his/her detailed comments. Below are the Reviewer’s comments followed by our responses.

  1. This study utilized NLP analysis to investigate the relationship between the unique abilities of adult people with ASD and the consequent adaptation of the working conditions for their benefit as well as for the benefit of the other employees. This topic is interesting and important. However, the writing is confusing, and the research methodology is not rigorous enough.

Reply: We highly appreciate this comment. The research methodology has been reorganized, and we hope it is now clearer to the Reviewer. The topic modeling and cloud tag techniques were used to evaluate which topics / prominent words appear in the scientific and the experts’ documents that investigate or are related to relationships between the unique abilities of people with ASD and the adaptation of the workplace’s environmental design to best fit their needs. (Refer to lines 757–777)

  1. The abbreviation “NLP” is used in the study title. It is better to use the full term “natural language processing” in the title.

Reply: Thank you for your comment. The NLP abbreviation has been replaced in the title by the full term. In the body of the text, the acronym appears initially in line 23 and is used thereafter for purposes of simplicity.

  1. Moreover, the authors mentioned “machine learning tools” in the title; but it seems only one text mining tool (Orange Toolkit) was used. Are there any other tools used?

Reply: We appreciate this comment. The reviewer is right. Only one tool was used (the OrangeTM Toolkit). This tool implements machine learning methods such as bag-of-words and cloud tags. Indeed, NLP itself is a subsection of machine learning. Therefore, the reference to machine learning has been obliterated from the title.

  1. The author used “people with autism” in some sentences (e.g., Line 41), “people with autism spectrum disorder” in Line 65, and “individuals with ASD” in Line 68 and 359. The term “autism” was changed to “autism spectrum disorder” by the American Psychiatric Association in 2013. The author should consistently use the term “autism spectrum disorder” or “ASD” in the manuscript.

Reply: We highly appreciate this comment. The term “autism” in conjunction with other words (such as people, individuals) has been replaced throughout the text with the term “ASD”. The term appears in its original form in the Methodology section (3.1), where it refers to words used in the scientific papers’ search (refer to lines 747–749).

  1. References are needed for Line 46-49.

Reply: Thank you for pointing this out. More references were added to the (original) lines 46–49 as suggested by the Reviewer.

  1. In the introduction, paragraph 4, the authors listed a partial list of jobs for people with ASD.However, I did not see references for each of the listed jobs. The authors should make it clear how this list was developed.

Reply: Thank you for pointing this out. The partial list of jobs was provided according to studies and recommendations appearing in the literature. We have added at least one reference for each of the listed job categories. An example of a source in this respect is the manuscript titled “Choosing the right job for people with autism or Asperger’s syndrome” by Grandin (1999).

  1. Similarly, the authors should explain how the workplace architectural design categories (mentioned in Line 76-132) were developed.

Reply: Thank you for your suggestion. The architectural design categories are based on the literature. Relevant references have been added to support the list of categories. (Refer to line 221.)

  1. The last paragraph of the introduction, Line 154-159, is unnecessary.

Reply: We thank you for your comment. The last paragraph of the Introduction has been deleted as suggested by the Reviewer.

  1. The authors proposed three “research questions” and constructed three “derived questions.” How the “research questions” and the “derived questions” are related? 8. Please explain why it is appropriate to include a review article (Tola et al. Line 259)? Will this contaminate the results?

Reply: We highly appreciate this comment. Indeed, there was need to explain better the connection between the research questions and the derived ones. Such explanation has been provided for each of the derived questions (refer to lines 577–589). Regarding the article by Tola et al. (line 259), our study not only supports its previous observations but also elaborates on the understanding that people with ASD may have physical as well as psychological needs. In addition, our study provides an implementation of Tola et al.’s findings as well as our own by means of architectural sketches. Finally, our study provides additional information derived from the NLP analysis of the expert’s corpus that elaborates on the existing knowledge and attitude in the explored domain.

  1. Citations are needed for the 10 papers in Table 2. It is better to include a summary of each paper’s main findings in Table 2.

Reply: Thank you for this comment. We have added citations and a summary of each paper’s main findings in Table 2.

  1. It is good to use the Google search engine as an additional literature searching source, but it is not rigorous enough to use it as a primary source; other databases need to be searched.

Reply: We highly appreciate this comment. As a literature search engine, Google Scholar is limited because it provides mostly abstracts and not full papers. For this reason, the Web of Science collection, which is based on prominent primary academic repositories, was used. However, we have taken up the Reviewer’s comment and added it as a suggestion for future research in the Limitations section. (Refer to lines 1453–1458.)

  1. How those categories mentioned in section 4.2 were derived from the literature?

Reply: Thank you for this question. The categories mentioned in 4.2 were derived from several references cited in the work of Cassidy (2018). In Section 5, the author provides possible prototypes of workplace environments that consider different types of architectural categories that may enhance the productivity, efficiency, and creativity of employees with ASD (refer to line 121).

  1. What contents in the identified literature were used for NLP analysis? The whole paper or only the paper’s results/findings. The data analysis method needs further description.

Reply: NLP was performed on the papers’ abstracts (refer to line 750). Following the Reviewer’s comment, Section 3.2 was elaborated further to clarify the topic modeling and word cloud analysis (refer to lines 757–777).

  1. The authors already listed many important ASD-related adjustments in the workplace’s architectural design in the introduction. What are the new findings in this study? They should be highlighted in the discussion.

Reply: Thank you for your suggestion. The new findings and their contribution to the existing literature are highlighted in the Discussion section as suggested by the Reviewer (refer to lines 1222–1236).

  1. There are a few formatting errors, such as Line 41-52, and Line 67.

Reply: Thank you for pointing these errors out to us. We have corrected them.

Reviewer 4 Report

Since universal design is more common used in special education, the literature review had better discussing the relationship with environmental design and included universal design as keyword in searching related journal articles.

In line 67-68 sentence disconnected.

Author Response

Reply to Reviewer 4's Comments

Revision of “How Well Environmental Design Is and Can Be Suited to People with Autism Spectrum Disorder (ASD): An NLP analysis”

(Ref: IJERPH, Manuscript ID -1657007)

We are grateful to Reviewer 4 for his/her careful reading of out paper and his/her detailed comments. Below are the Reviewer’s comments followed by our responses.

  1. Since universal design is more common used in special education, the literature review had better discussing the relationship with environmental design and included universal design as keyword in searching related journal articles

Reply: We highly appreciate this comment. According to the Reviewer’s suggestion, a search in the WoS repository was performed with the keywords “Autism”, “Workplace”, and “Universal Design”. The methodology was updated accordingly in Section 3.1 and the results were reported in Section 4.1 (refer to lines 745–747 and 907–911).

  1. In line 67-68 sentence disconnected.

Reply: Thank you for pointing this out. The sentence was rewritten as required.
